# High-Quality Resolution of the Outbreak-Related Zika Virus Genome and Discovery of New Viruses Using Ion Torrent-Based Metatranscriptomics

**DOI:** 10.3390/v12070782

**Published:** 2020-07-21

**Authors:** Silvia I. Sardi, Rejane H. Carvalho, Luis G. C. Pacheco, João P. P. d. Almeida, Emilia M. M. d. A. Belitardo, Carina S. Pinheiro, Gúbio S. Campos, Eric R. G. R. Aguiar

**Affiliations:** 1Laboratory of Virology, Instituto de Ciências da Saúde, Universidade Federal da Bahia, Salvador, Bahia 40.110-100, Brazil; sissardi@yahoo.com.br (S.I.S.); hughescv@gmail.com (R.H.C.); gubiosoares@gmail.com (G.S.C.); 2Post-Graduate Program in Biotechnology, Instituto de Ciências da Saúde, Universidade Federal da Bahia, Salvador, Bahia 40.110-100, Brazil; lgcpacheco@gmail.com (L.G.C.P.); carinasilvapinheiro@gmail.com (C.S.P.); 3Department of Biochemistry and Immunology, Instituto de Ciências Biológicas, Universidade Federal de Minas Gerais, Belo Horizonte (UFMG), Minas Gerais 31270-901, Brazil; jpereiradealmeida.mg32@gmail.com; 4Post-Graduate Program in Immunology, Instituto de Ciências da Saúde, Universidade Federal da Bahia, Salvador, Bahia 40.110-100, Brazil; emiliammabelitardo@gmail.com; 5Virus Bioinformatics Laboratory, Department of Biological Science (DCB), Center of Biotechnology and Genetics (CBG), State University of Santa Cruz (UESC), Rodovia Ilhéus-Itabuna km 16, Ilhéus, Bahia 45652-900, Brazil

**Keywords:** virus identification, *Zika virus*, RNA deep sequencing, Ion Torrent, metatranscriptomics

## Abstract

Arboviruses, including the *Zika virus*, have recently emerged as one of the most important threats to human health. The use of metagenomics-based approaches has already proven valuable to aid surveillance of arboviral infections, and the ability to reconstruct complete viral genomes from metatranscriptomics data is key to the development of new control strategies for these diseases. Herein, we used RNA-based metatranscriptomics associated with Ion Torrent deep sequencing to allow for the high-quality reconstitution of an outbreak-related *Zika virus* (ZIKV) genome (10,739 nt), with extended 5′-UTR and 3′-UTR regions, using a newly-implemented bioinformatics approach. Besides allowing for the assembly of one of the largest complete ZIKV genomes to date, our strategy also yielded high-quality complete genomes of two arthropod-infecting viruses co-infecting C6/36 cell lines, namely: *Alphamesonivirus 1 strain Salvador* (20,194 nt) and *Aedes albopictus totivirus-like* (4618 nt); the latter likely represents a new viral species. Altogether, our results demonstrate that our bioinformatics approach associated with Ion Torrent sequencing allows for the high-quality reconstruction of known and unknown viral genomes, overcoming the main limitation of RNA deep sequencing for virus identification.

## 1. Introduction

Arthropode-borne viral infections (arboviroses), transmitted to mammals by hematophagous arthropod vectors, have recently emerged as one of the most important threats to human health. Due to their worldwide occurrence and fast adaptation to environmental changes, mosquitoes from the *Aedes* genus play an important role in the transmission of viral etiological agents of emerging human infections, including *Dengue virus* (DENV), *Zika virus* (ZIKV), *Chikungunya virus* (CHYKV) and *Yellow fever virus* (YFV) [1,2]. Many epidemic events have been reported, which are associated with the lack of efficient vaccines, the rapid evolution of viral genomes, and inefficient control strategies. One of the last episodes was the Zika outbreak in Brazil in the period 2015–2016, which impacted more than 200,000 individuals by the end of 2016 [3]. The impact of this outbreak was amplified due to the *Zika virus* infection’s relationship to many neurological disorders, such as Guillain–Barre and microcephaly syndromes, which led the World Health Organization to declare the Brazilian Zika outbreak as one of International Concern by 2016. Different from African lineage, the Asian lineage of *Zika virus*, supposedly the origin of the strains circulating in Brazil, exhibits a different profile with lower pathogenicity and prolonged infections [4]. These differences have been attributed to mutations in the viral genome, mainly in untranslated regions [5,6]. The long-term infection, which could facilitate mosquito transmission, associated with the high susceptibility of the population likely explains the rapid spread of the virus that reached almost all Brazilian states in only a few months, different, for example, to the *Dengue virus* whose spread took decades [7].

In addition to arboviruses, *Aedes* mosquitoes are also known to carry a vast range of viruses, most of them insect-specific [8,9,10]. Although they do not cause direct harm to human health, they have been described as impacting mosquito susceptibility to other viral infections and vector competence [11,12,13]. Therefore, the knowledge of the viruses circulating in mosquitoes (virome) is important in understanding mosquito–arbovirus interactions as well as in developing new strategies of control, and in the prevention of epidemic events. In this context, metagenomics associated with deep sequencing dramatically increased the resolution of viral genomes and the identification of new viruses [14,15]. However, while DNA sequencing restricts the identification of viruses that use DNA intermediates in the replication cycle, RNA sequencing allows for the identification of all classes of viruses, since almost all of them generate RNA intermediates during their replication. Indeed, metatranscriptomics has been successfully applied to the identification of RNA and DNA viruses in a vast range of organisms, such as plants, insects, fungi, and mammals [8,16,17,18]. Despite being a powerful strategy, there are technical limitations that hamper confident identification of whole viral genomes, such as length of sequenced reads, repetitive genomes, differential expression of virus segments, and natural variation in the virus population that make the confident assembly of contiguous sequences a challenging task [19,20]. Here, we applied RNA-based metatranscriptomics associated with Ion Torrent deep sequencing and a newly developed Bioinformatics approach to the high-quality reconstitution of viral genomes. Using this strategy, we successfully reconstituted an outbreak-related *Zika virus* genome with extended 5′-UTR and 3′-UTR regions and the genomes of two arthropod-infecting viruses co-infecting C6/36 cell lines. Altogether, our data demonstrate that our bioinformatics approach associated with Ion Torrent sequencing allows for high-quality reconstruction of known and unknown viral genomes.

## 2. Material and Methods

### 2.1. Cell Culture, Infection, Library Preparation and Sequencing

C6/36 cells (*Aedes albopictus*) were maintained in Leibovitz’s L15 medium (Gibco^®^, CA, USA), supplied with 5% Fetal Bovine Serum (FBS) (Gibco^®^), 10% Tryptophan Broth (Sigma-Aldrich^®^, Boston, MA, USA) at 28 °C. The cells were inoculated with a *Zika virus* strain (partial fragment of E protein deposited at GenBank KR816334 from Zika virus isolate BR/UFBA/LabViro/Ex1—see Figure 1b) derived from a serum sample collected from a patient in the Brazilian outbreak of 2015 with MOI 1, and after observation of the cytophatic effect (4 days), the supernatant of the infected cells was centrifugated (10.000 g at 10 °C for 20 min) and then submitted to ultracentrifugation (100.000 g at 4 °C for 2.5 h). The pellet was resuspended in phosphate saline solution (PBS) and used to perform RNA extraction through the QIAamp Viral RNA Mini Kit (QIAGEN, Hilden, Germany). Quantification of the total extracted RNA was performed by fluorimetry using Qubit equipment (Life Technologies, Carlsbad, CA, USA). The RNA library was constructed from the extracted RNA using Ion Total RNA-seq kit v2 according to the manufacturer’s protocols. Sequencing was performed using an Ion 540™ Chip Kit (Life Technologies) in the Ion OneTouch™ 2 system and the Ion S5 instrument. Sequencing data were deposited at the SRA database under accession number: SRR11591975.

### 2.2. Reference Genomes.

*A. albopictus* genome was downloaded from VectorBase (www.vectorbase.org). All complete bacterial genomes used to remove possible contamination in sequenced reads were downloaded from the RefSeq database [21]. Sequence and metadata of the *Zika virus* were downloaded from Virus Pathogen Resource (ViPR) [22], and only complete genomes were analyzed. For phylogenetic analysis, top Blast hits for each new viral contig were retrieved from the NCBI database ordered by score.

### 2.3. Bioinformatics Analyses.

Pre-processing: Reads derived from RNA deep sequencing were submitted to quality checking and filtering using FastQC [23] and FastX toolkit [24]; reads with an average Phred quality below 20 were discarded. To enrich the viral sequences, filtered reads were aligned against the *A. albopictus* genome, and bacterial genomes present on the RefSeq database using Bowtie2 [25]. Only unaligned reads were used for the subsequent analysis. Assembly: Assemblies were performed using SPAdes assembler [26] with the following variations: (1) standard: Filtered reads were used as input and standard parameters kept; (2) meta: Filtered reads were analyzed as derived from the metagenomics sample, using the parameter “--meta”; and (3) meta (trusted) + regular: Viral contigs assembled in “meta” strategy were used as trustable contigs, the parameter “--trusted-contigs,” together with filtered reads. All strategies were run with the parameter “—iontorrent.” In strategies 1 and 2, the parameter “--cov-cutoff” was set to “auto.” In strategy 3, “--cov-cutoff” was set to “off.” All contigs with significant hit with viral sequences were submitted to manual curation. Manual curation of assembled genomes: Assembled viral genomes were curated based on structure (presence of ORFs, conserved domains, secondary structure) and RNA coverage (evaluation of continuous coverage along segment). The viral genome sequences produced in this work were deposited on GenBank under accession numbers MN101548-MN101550. Characterization of viral contigs: Assembled contigs were submitted to sequence similarity searches using Blast software [27] against non-redundant NCBI databases of nucleotide (NT) and aminoacids (NR), requiring minimum e-values of 1e^−5^ and 1e^−3^, respectively. Contigs that showed significant similarity to viral sequences were further analyzed. Conserved domains were analyzed using the HMMER tool [28] with the Pfam database. ORF prediction: The prediction of the open read frame of the assembled sequences was performed using the NCBI ORF finder (https://www.ncbi.nlm.nih.gov/orffinder/) with “standard” genetic code defining minimum ORF length to 100 nt. RNA coverage analysis: Reads were re-aligned to assembled viral genomes with Bowtie2; SAM files were processed and quantified using SAMtools [29] and BEDTools [30], and for visual inspection of reads coverage we used IGV [31]. Secondary Structure Analysis: Analyses of the secondary structure of 5′ and 3′ UTR of the Zika virus genomes were performed using the RNAfold webserver from the Vienna platform [32] with standard parameters. Phylogenetic and Genotypic analysis: Phylogenetic analysis was performed using MEGA software [33]. Maximum likelihood phylogenetic trees were constructed based on the whole viral genome with the Kimura 2-parameter method, or RdRp protein using the Poisson substitution model for *Alphamesonivirus* and *Totivirus-like virus*, respectively. The trees were drawn with branch sizes measured in the number of substitutions per site. The percentage of trees in which the associated taxa clustered together in the bootstrap test (1000 replicates) is shown next to the branches. Investigation of the *Zika virus* genotype was performed using well defined phylogenetic clusters with information of geographic region integrated in the *Zika virus* Typing Tool [34]. RNA coverage analysis: Preprocessed reads were compared to each reference sequence using Bowtie2, allowing a maximum of two mismatches. Bowtie2’s SAM output was used to compute per base coverage with the BEDTools package. Per base coverage was plotted using R language [35] with the ggplot2 package [36].

### 2.4. Validation of Totivirus presence by RT-PCR and Sanger Sequencing

Total RNA from C6/36 culture cells was extracted using QIamp^®^ RNA MiniKit and submitted to one-step RT-PCR using the following primers (F: ACGTGCCAGCTGTGTCTATG and R: ACACCAGCCATCAAGGACG) and the Access RT-PCR System (Promega, MA USA). Cycling conditions were defined as follows: 45 °C for 45 min, 95 °C for 2 min, and 35 cycles of 95 °C for 30 s, 58 °C for 30 s, and 72 °C for 1 min, the final extension step was at 72 °C for 5 min. The primers amplified a fragment of 739 bp when visualized in 2% agarose gel, stained with ethidium bromide and detected by an ultraviolet-transilluminator system. RT-PCR product was cleaned up using the QIAquick PCR Purification Kit, according to the manufacturer’s instructions (Qiagen, Germany). Sequencing was performed by the 3500 Genetic Analyzer using the big dye terminator method, and it was carried out in 60 ng of purified RT-PCR product and 4.5 pmol of the same primers used for the RT-PCR.

## 3. Results and Discussion

RNA deep sequencing produced 20,499,807 reads. After preprocessing, 9,593,282 were kept for posterior analysis. Since the goal was to reconstitute the *Zika virus* genome that only has a small fragment of the protein E available, we discarded all sequences derived from mosquito genome (1,734,635). The remaining reads (7,858,647) were used to perform assembly using different strategies described in Figure 1a, in the top panel. We first tried to use SPAdes with the default parameters (highlighted in gold in Figure 1a) referred here as ‘regular.’ Although we identified many contigs derived from the *Zika virus*, we were not able to reconstitute the viral genome as a single segment. Since our data are derived from deep sequencing of all RNAs present in the cell, we performed the assembly using the “--*meta*,” a variant of the assembler optimized for metagenomic data. Using this strategy, we considerably improved the assembly of the *Zika virus* genome, reducing the number of contigs from 15 to 4, and increasing size average from 569.1 to 2475 nt (highlighted in dark blue in Figure 1a). Although we assembled longer contigs, the genome was still fragmented. Interestingly, we observed that fragmentation in the Zika genome correlated with the positions that had high variability in the density of coverage (Figure 1a—bottom panel). This is consistent with the literature showing that differences in coverage of overlapping sequences can lead to the split of supposedly contiguous sequences into different contigs [19]. This is a problem mainly when assembling viral genomes from RNAs since viral genes can show different transcription profiles [37,38]. To overcome this limitation, we decided to use contigs derived from the “meta” strategy as trusted contigs trying to extend these contiguous sequences. This new strategy, which is hereafter referred to as “meta (trusted) + regular,” resulted in the complete reconstitution of the *Zika virus* genome containing 10,739 nt (Figure 1a,b—top panel). An overview of the final approach is shown in Appendix A and described in detail in the Material and Methods section.

It is important to highlight that this viral genome extended in 214 bp (97 bp in 5′-UTR and 117 bp in 3′-UTR including coding regions) is the closest viral sequence derived from the 2015 outbreak, the Zika virus strain Bahia 09, that shows ~98 % of coverage with ~99% of identity (Figure 1b) [3]. Since confident assembly of UTR regions is challenging due to the lower number of reads in comparison to non-UTR regions, we further manually cured these regions by investigating aligned reads using a genome browser. As expected, we noted that UTR regions presented a considerably lower number of reads in comparison to non-UTR regions, 257,690 compared to 3,148,506, respectively. However, although the number of reads was restricted, we observed concordantly alignments in both UTR regions, 6784 and 250,906 for 5′ and 3′ extremities, respectively (Appendix A). Furthermore, analysis of the UTR region sequences suggests that both of them form secondary structures, which has been shown previously to be essential to the virus driving viral fitness and pathogenesis (Figure 2) [5]. This demonstrates that our strategy based on Ion Torrent metatranscriptomics associated with de novo assembly is able to reconstitute whole viral genomes. Indeed, performing a global overview of complete *Zika virus* genomes, it is noticeable that our newly assembled genome figures are among the top 15 in length out of more than 800 viral sequences available, which seems to be independent of collection year or country (Figure 1c). Indeed, recent work has shown that strategies based on amplicon are still preferentially used to reconstruct ZIKV genomes, which create a bias to the coding region since amplicons are mostly designed to anchor at genic regions with lower mutation rates [39].

Similarly, for the previous Zika virus strains derived from the outbreak in 2015, the viral strain assembled in this work clustered together with other strains with Asian genotype (Figure 1d). Comparative analysis among all complete sequences deposited at the VIPBRC database revealed that differences among viral genomes are mainly in untranslated regions, which is consistent with the literature [4,6,39]. Curiously, untranslated regions have been linked to differences in virulence of different arboviruses [4,40,41]. In addition, these regions are also suggested to be involved in other viral characteristics, such as host preference and tissue tropism [4,6,42]. This result highlights the need for achieving complete viral genomes to better our understanding of virus–host interactions.

Notably, besides the identification of contigs showing sequence similarity to the *Zika virus*, with the new assembly strategy, we were also able to reconstruct two large contigs showing similarities to viral sequences. One contig of 20,194 nt showed a sequence similarity at the nucleotide level (99% identity with e-value = 0) with viruses from *Alphamesonivirus* genus while one contig with 4618 nt presented a similarity at the protein level (~40% identity with e-value = 1e-150) with viruses from the *Totiviridae* family, which likely represents a new viral species. Phylogenetic analysis based on the nucleotide sequence revealed that the Alphamesonivirus identified in C6/36 cell lines is a new strain of a previously identified *Alphamesonivirus 1* from the *Mesoniviridae* family, a group of single-stranded RNA viruses that have been described in mosquitoes (Figure 3a) [43]. This is the first report of this viral family in *A. albopictus* mosquitoes. It was named *Alphamesonivirus 1 strain Salvador*. Phylogenetic analysis based on RdRp protein of contig related to the *Totiviridae* family suggested its relationship to a newly proposed family, *Totivirus-like*, which is different from totiviruses that have Fungi as natural hosts, and have been described in a vast range of organisms (Figure 3b) [44,45,46,47,48]. This new virus was named *Aedes albopictus totivirus-like*.

For both of the two new viruses identified, we observed an ORF structure consistent with their closely related viruses (Figure 3c,d). In addition, we observed a considerable number of reads aligned with the *Totivirus* and *Alphamensonivirus* assembled genomes, 16,319 and 40,521 respectively, representing 0.09% and 0.22% of the library. Furthermore, RNA density along viral genomes showed continuous coverage, suggesting that they were correctly assembled (Figure 3c,d). To confirm the detection of the newly identified *Totivirus-like* sequence, experimental validation was performed by RT-PCR, and we were able to detect the viral genome in the same C6/36 population we used to perform deep sequencing. We also retrieved a fragment of its genome through Sanger sequencing in order to compare with the contiguous sequence obtained through metatranscriptomics (Appendix A). We observed a high similarity between the fragment retrieved through Sanger sequencing and the assembled using deep sequencing (99% identity), suggesting the high accuracy of our metatranscriptomics strategy to reconstitute viral genomes.

## 4. Conclusions

Altogether, our results indicate that our strategy based on RNA deep sequencing using Ion Torrent technology allows the confident reconstitution of known and unknown viruses. Using this strategy, we were able to reconstruct the complete genome of the outbreak-related *Zika virus* genome, including 5′ and 3′ UTR regions, that would not be possible using standard approaches. Furthermore, we also assembled the full genomes of two unknown viruses co-infecting the stock, *Alphamesonivirus* and the newly described *Totivirus-like*, highlighting the potential of our strategy for monitoring viral contaminations. Therefore, our work highlights the importance of the development of new unbiased metagenomic approaches for the identification of viral sequences.

## Figures and Tables

**Figure 1 viruses-12-00782-f001:**
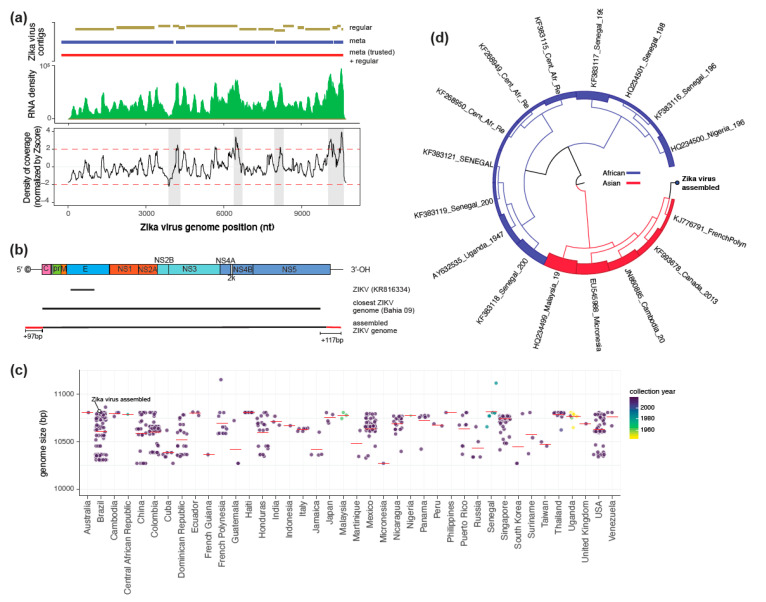
Significant improvement of the *Zika virus* genomic sequence identified in the 2015 outbreak in Bahia-Brazil. (**a**) Overview of assembly strategies and RNA coverage of the *Zika virus* genome. For each of the assembly strategies, contigs larger than 500nt were represented as continuous vertical bars. The density pattern was computed by normalizing the per-base coverage by Z-score. Genomic regions in which the Z-score of RNA coverage was greater than 2 were highlighted in gray. (**b**) Comparative analysis of the coverage of the *Zika virus* genome considering the fragment available at GenBank, the closest ZIKV genome (Bahia 09), and a new version of the *Zika virus* genome assembled in this work. (**c**) Comparative analysis of 824 complete *Zika virus* genomes deposited on the VIPBRC platform in comparison to the new version of the genome of the *Zika virus* (KR816334) reconstituted in this work. Horizontal red lines represent the average size of the genomes from each country. (**d**) Analysis of the *Zika virus* genotype on the Zika Virus Typing Tool [34].

**Figure 2 viruses-12-00782-f002:**
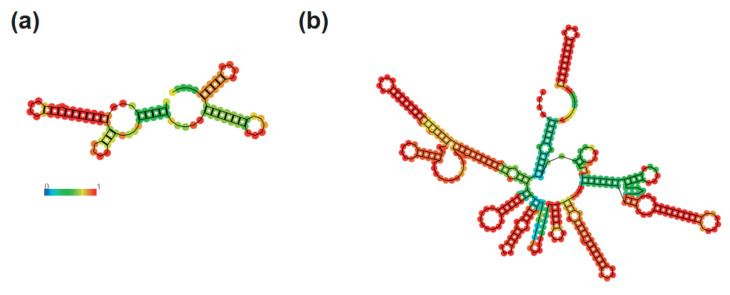
Secondary structure of the untranslated regions from the *Zika virus* genome assembled. Analysis of the secondary structure of (**a**) 3′ and (**b**) 5′ UTR of Zika virus genome.

**Figure 3 viruses-12-00782-f003:**
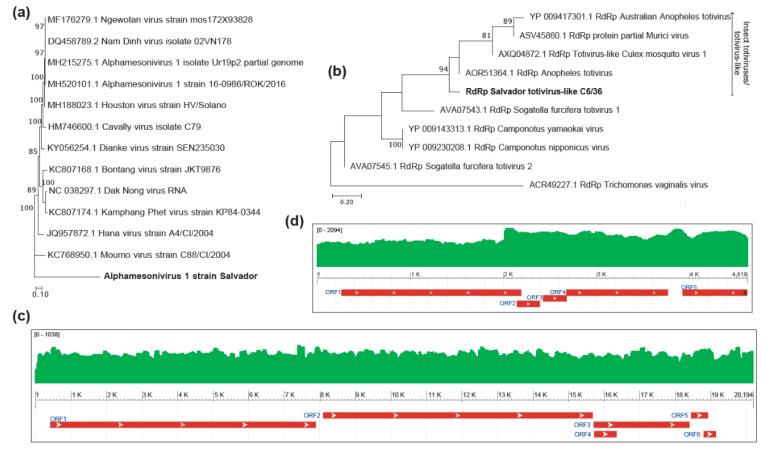
Characterization of viruses identified in the *A. albopictus* C6/36 cell line. Maximum likelihood phylogenetic trees of the (**a**) new strain of *Alphamensonivirus* and (**b**) new viral specie related to the *Totiviridae* family. RNA coverage of identified viral genomes are shown for (**c**) *Alphamensonivirus* and (**d**) *Totivirus-like*.

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
