# Peer review of "High-Quality Resolution of the Outbreak-Related Zika Virus Genome and Discovery of New Viruses Using Ion Torrent-Based Metatranscriptomics"

_viruses, 2020, doi:10.3390/v12070782_

Round 1
Reviewer 1 Report
In this report, an unbiased metagenomic sequencing approach and assembly strategy for RNA viruses is presented. The authors have infected C6/36 cells from Aedes albopictus with a Zika virus strain and then performed RNA sequencing with Ion sequencing equipment of total RNA. Their approach allowed direct assembly of the full genome, including the 5’ and 3’ untranslated regions. Two additional viral genomes were identified and completely assembled from the same sample, one of them corresponding to a novel insect virus species.
The results show that this is a feasible approach for sequencing and assembly of RNA virus samples that may be of use to other researchers.
Overall, the data are clearly presented, although some the authors might wish to consider some modifications:
- Please include a description of the experimental design (how the cells were infected, with what strain and how the sample was prepared for RNA sequencing) in the Results and Discussion section. In particular, it is not immediately evident if the full length sequence of the Zika virus strain used to infect these cells was available before (from the Genbank accession number provided apparently this is not the case). Please extend or rephrase lines 164-166 to make this result clear, providing length of fully assembled genome, complete strain name, degree of identity to the Bahia 09 strain along the description of UTR lengths.
- In the Methods section, please describe the infection and sample preparation procedure with some detail. In particular, multiplicity of infection, time or criteria for harvest and method for preparation. Please see line 79: are “frozen cultured cells” infected cells?
- Please provide the total number and fraction of reads that were incorporated into the final assembled Zika virus genome sequence, along with coverage of the UTRs as opposed to non-UTR regions in the text.
- Two new insect viruses are identified which are mentioned to coinfect C6/36 cell lines (line 29) in the abstract. However, this seems to correspond to a single observation from this specific sample and therefore it would be good to clarify whether this is a possible contamination event or why the authors think that this may be a property of this cell line. Have the authors checked different C6/36 batches for the presence of these viruses? Please include information about the number and fraction of reads incorporated into these viral sequences.
- The Figure 2 does not seem to be referred to in the text, If the figure is kept, also please include an indiction as to how secondary structure was computed.
- Please describe in the legend what the bars on panel c of Figure 1 indicate.
- Regarding supplementary information, please keep only figures 1 and 2 as methods are a repeat of the methods in the manuscript and Table S1 does not correspond to the description in the text (remove mention in the main text or add correct table). The oligonucleotide sequences from this Table S1 can be included in Figure S2 or main methods section. In Figure S2 panel A, please label or indicate nature of lanes.
- Please rewrite lines 40-42 in a single non-redundant sentence.
- Please remove the ISV abbreviation on line 60 as it is not standard and not further used.
- Please modify the end of the sentence on lines 73-74 “that make it difficult the confident assembly of contiguous sequences” to “that make the confident assembly of contiguous sequences difficult”.
- On line 213 please modify “achieved” for “obtained”.
Author Response
We thank the referee for the careful evaluation of our manuscript. Please find below the Point-by-point response to the questions raised by Reviewer 1:
- Please include a description of the experimental design (how the cells were infected, with what strain and how the sample was prepared for RNA sequencing) in the Results and Discussion section. In particular, it is not immediately evident if the full length sequence of the Zika virus strain used to infect these cells was available before (from the Genbank accession number provided apparently this is not the case). Please extend or rephrase lines 164-166 to make this result clear, providing length of fully assembled genome, complete strain name, degree of identity to the Bahia 09 strain along the description of UTR lengths.
Answer:
We apologize to the referee for the lack of clarity. We have updated the Material and Methods section, sub-topic “Cell culture, Infection, Library preparation and Sequencing” with more detailed information about the experimental design (currently lines 81-89). We also made it clear on the Material and Methods and Results sections and Figure 1 legend that only a small fragment of the protein E of the Zika virus that we used to inoculate the C6/36 cell lines was available (currently lines 150-151, 177-182). In addition, we included a statement at Results section detailing the size of the full genome and complete comparison to Zika virus Bahia 09 (lines 183-185).
- In the Methods section, please describe the infection and sample preparation procedure with some detail. In particular, multiplicity of infection, time or criteria for harvest and method for preparation. Please see line 79: are “frozen cultured cells” infected cells?
Answer:
We apologize to the referee for the lack of details. We have now re-written the sub-topic “Cell culture, Infection, Library preparation and Sequencing” sup-topic of Material and Methods section to provide detailed information about infection and sample preparation (currently lines 81-89).
- Please provide the total number and fraction of reads that were incorporated into the final assembled Zika virus genome sequence, along with coverage of the UTRs as opposed to non-UTR regions in the text.
Answer:
We have included in the Result section detailed information about number of reads aligned to complete Zika virus genome assembled as well as the information relative to UTR and coding regions discriminated by origin (currently lines 183-192).
- Two new insect viruses are identified which are mentioned to coinfect C6/36 cell lines (line 29) in the abstract. However, this seems to correspond to a single observation from this specific sample and therefore it would be good to clarify whether this is a possible contamination event or why the authors think that this may be a property of this cell line. Have the authors checked different C6/36 batches for the presence of these viruses? Please include information about the number and fraction of reads incorporated into these viral sequences.
Answer:
We Agree with the referee. Indeed, initially it was difficult to determine whether the virus could be originated from ZIKV stock, the C6/36 cell culture or contamination. However, the samples used on the RT-PCR were derived from the C6/36 stock, which means that the virus are likely present on cell culture. In addition, viral families which the viruses identified are related have been described infecting different insects. In addition, persistent silent viral infection are commonly found in insect cell stocks, including C6/36 cells (Bolling et al, 2012). Therefore, We believe the viruses are probably infecting C6/36 stock instead of originated from external contamination.
We have added on the Results section the information regarding the number and fraction of reads aligned to the genomes of the identified viruses (lines 233-235). We observed a considerable number of reads derived from Totivirus and Alphamensonivirus genomes, 16,319 and 40,521 representing 0.09% and 0.22% of the library, respectively.
Bolling, B.G.; Olea-Popelka, F.J.; Eisen, L.; Moore, C.G.; Blair, C.D. Transmission dynamics of an insect-specific flavivirus in a naturally infected Culex pipiens laboratory colony and effects of co-infection on vector competence for West Nile virus. Virology 2012, 427, 90–97, doi:10.1016/j.virol.2012.02.016.
- The Figure 2 does not seem to be referred to in the text, If the figure is kept, also please include an indiction as to how secondary structure was computed.
Answer:
The Figure 2 was already cited in the Result section, on the line 169 of submitted version and currently on the line 193 of the revised version. We also included in the Material and Methods section the Zika virus genome (lines 122-124).
- Please describe in the legend what the bars on panel c of Figure 1 indicate.
Answer:
We apologize to the referee. We have updated the Figure 1 legend to make it clearer. The horizontal bar in the panel (c) represent average size of the genomes derived from each country (line 181).
- Regarding supplementary information, please keep only figures 1 and 2 as methods are a repeat of the methods in the manuscript and Table S1 does not correspond to the description in the text (remove mention in the main text or add correct table). The oligonucleotide sequences from this Table S1 can be included in Figure S2 or main methods section. In Figure S2 panel A, please label or indicate nature of lanes.
Answer:
We apologize to the referee. The supplementary information was updated according to the reviewer comments to include the origin of the lanes (Supplementary Figure 3 legend). We also included the oligonucleotides used on the main Material and Methods section to avoid unnecessary table (currently on the lines 137-138).
- Please rewrite lines 40-42 in a single non-redundant sentence.
Answer:
We apologize to the referee. We have rewritten the sentence to remove redundancy ( currently on the line 40-41).
- Please remove the ISV abbreviation on line 60 as it is not standard and not further used.
Answer:
We agree with the referee. We have removed the abbreviation ( currently on the line 59 ).
- Please modify the end of the sentence on lines 73-74 “that make it difficult the confident assembly of contiguous sequences” to “that make the confident assembly of contiguous sequences difficult”.
Answer:
We agree with the referee. We have modified the sentence according to the suggestion of the referee ( currently on the lines 72-73).
- On line 213 please modify “achieved” for “obtained”.
Answer:
We agree with the referee. We have replaced the word (currently on the line 240).
Reviewer 2 Report
Sardi et al. reported a method to obtain the high-quality reconstitution of an outbreak-related Zika virus (ZIKV) genome with completed 5’-UTR and 3’-UTR regions. They discovered an extra 466 bp at both ends (97 bp in 5’-UTR and 369 bp in 3’-UTR including coding regions). They achieved this by using Ion Torrent sequencing in combination with a newly developed bioinformatics approach.
Given the important roles of both 5’ and 3’-UTR in regulating viral genome replication and protein translation, it is critical for the field to get the complete sequences of both UTR regions for the both ends. In this sense, the authors provided a reliable method to help better understanding viral pathogenicity.
Specific point:
Due to the variable quality of viral RNA templates and sequencing techniques used, it is difficult to obtain the full length of UTR. It is important that authors will provide another method to confirm their findings, such as 5’ and 3’ RACE.
Author Response
We thank the referee for the careful evaluation of our manuscript. Please find below the Point-by-point response to the specific question raised by Reviewer 2:
Specific point:
Due to the variable quality of viral RNA templates and sequencing techniques used, it is difficult to obtain the full length of UTR. It is important that authors will provide another method to confirm their findings, such as 5’ and 3’ RACE.
Answer:
We agree with the referee that obtain the full length of UTR sequences is a hard work. However, we add a supplementary Figure S2 showing the long RNA reads aligned to 5’ and 3’ UTR regions, demonstrating that there is a considerable number of concordant reads. It is also important to highlight that we have related sequences available in public databases, which showed high concordance with our assembled genome (98% of coverage with 99% of identity), which is expected for different strains of the same virus. Furthermore, the 5’and 3’ UTR secondary structure were consistent with which has been described in the literature. We also include a discussion about the limitation of assembling UTR regions of viral genomes at the Result section (lines 183-191). In summary, with the high read coverage, concordant reads spanning both UTR and non-UTR regions, and coherent sequence similarity with public sequences, we are confident to have presented strong pieces of evidence of a reliable assembled genome, including UTR regions.